# Correlates of bullying victimization among school adolescents in Nepal: Findings from 2015 Global School-Based Student Health Survey Nepal

**Tamanna Neupane** [1]*, **Achyut Raj Pandey** [2], **Bihungum Bista** [1], **Binaya Chalise** [3]

1 Nepal Health Research Council, Ramshah Path, Kathmandu, Nepal, 2 Abt Associates, Lalitpur, Nepal, 3 Graduate School for International Development and Cooperation, Hiroshima University, Hiroshima, Japan

* tamanna.neupane@gmail.com

## Abstract

### Background

Bullying is an emerging risk factor for poor mental health outcomes adversely affecting children and adolescents. However, it has rarely caught the attention of the health and education sector due to lack of evidence in many countries including Nepal. This study aimed to assess the prevalence and factors associated with bullying behavior among adolescent students in Nepal.

### Methods

We used nationally representative data from the Nepal Global School-Based Student Health Survey that involved two-stage cluster sampling design with the use of a standard set of self-administered questionnaires. Complex sample analysis was done to determine the prevalence and correlates of bullying among 6529 students of 68 schools studying in grade 7 to 11 using descriptive analysis and multivariable logistic regression.

### Results

The overall prevalence of bullying among Nepalese school adolescents was 51% (55.67% in male and 46.17% in female). Bullied adolescents more commonly reported mental health problems with higher risk of loneliness (aOR 1.36, 95% CI: 1.12, 1.64), anxiety (aOR 2.04, 95% CI: 1.65, 2.52), suicide attempt (aOR 2.08, 95% CI: 1.54, 2.81), school absenteeism due to fear (aOR 1.72, 95% CI: 1.34, 2.21) and school truancy (aOR 1.48, 95% CI: 1.17, 1.87). A significant association was seen between bullying victimization and negative health behaviors like involvement in physical fights (aOR 3.64, 95% CI: 2.94, 4.51) and tobacco use (aOR 2.05, 95% CI: 1.15, 3.65).

### Conclusion

School bullying is significantly associated with mental health factors like loneliness, anxiety, suicide attempt, school absenteeism and risky behavioral factors like smokeless tobacco

**Data Availability Statement:** The dataset supporting the conclusions of this article is open access data and is available in the WHO website:

https://www.who.int/ncds/surveillance/gshs/
nepaldataset/en/.

**Funding:** Abt Associates Pty Ltd (Nepal Office)
provided support in the form of salary for author
Achyut Raj Pandey and he did not have any
additional role in the study design, data collection
and analysis, decision to publish or preparation of
the manuscript. The specific roles of authors are
articulated in the authors' contributions, section.

**Competing interests:** Abt Associates Pty Ltd
(Nepal Office) provided support in the form of
salary for author Achyut Raj Pandey and he did not
have any additional role in the study design, data
collection and analysis, decision to publish or
preparation of the manuscript. The specific roles of
authors are articulated in the authors'
contributions, section. Also, there are no patents,
products in development or marketed products
associated with this research to declare. This does
not alter our adherence to PLOS ONE policies on
sharing data and materials.

use and involvement in physical fight. The insights provided by these findings have important implications for planning anti-bullying strategies in school settings in the Nepalese context.

## Introduction

Bullying is a global public health priority, with negative impacts on the health and education of children and adolescents [1, 2]. It is defined as repeated aggressive behavior and actions between victims and perpetrators due to the imbalance of power [2, 3]. In five South Asian countries, (Bangladesh, Bhutan, India, Nepal and Pakistan), a total of 7, 68,238 Disability Adjusted Life Years (DALYs) were attributable to childhood bullying in 2017. It was an increase of 90% from 1990 when 403987 DALYS were attributable to bullying. Approximately, 5,42,497 DALYs were attributable to childhood bullying in low Socio-demographic Index (SDI) countries in 2017, which is an increase of 150% from 1990 (total 2,16,786 DALYs). Similar to South Asia and Low SDI countries, DALYs attributable to childhood bullying seems to be in increasing trend in Nepal. In 2017, 17,324 DALYs were attributable to childhood bullying in Nepal, which is an increase from 8,609 DALY in 1990. Almost 32.69 DALYs per 100,000 from anxiety disorder and 25.27 DALYs per 100,000 from depressive disorder are attributable to bullying in Nepal [4].

School bullying is one of the pressing issues faced by adolescents in educational settings. Existing research has revealed that bullied adolescents are at increased risk of poor physical [5–7] and mental health problems (such as anxiety, depression, psychiatric disorder), with immediate to long term health impacts [8–10]. Bullying can even lead victims to self-harm thoughts and suicidal behavior [11, 12]. While bullying is a risk factor of mental health, it may also be a consequence of poor mental health. Studies suggest bullying victimization as a predictor of mental health outcomes and vice-versa (mental health outcomes as a predictor of bullying). The direction of this reciprocal relationship is, however, contingent on the gender and maturity of adolescents [13, 14].

Victimized students fear to be at school and are more likely to miss school, play truant, avoid school activities and show poor academic performance [15–17]. Results from these studies have highlighted the importance of promoting a safe and bully-free learning environment for better educational development of students. Some research also revealed that victimization is related to the use of harmful substances like cigarette, tobacco, alcohol [18–20] as well as involvement in sexual risk behavior [21] and physical fight [22, 23]. There is emerging evidence around the association of bullying on body image. For instance, underweight or overweight adolescents were more likely to be bullied compared to healthy weight adolescents [24, 25].

Though, bullying is receiving increased attention in many countries these days, it is still the unexplored area in most of the developing countries. Some previous attempts have identified school bullying as a critical public health issue in Nepal [26–28]. These studies have investigated the association of bullying with few of the variables like age, ethnicity of participants, school type, depression and psychosomatic symptoms. However, we found that this topic still lacks in-depth evidence and is one of the under-researched areas in Nepal. This study reviewed Nepal Global School Based Student Health Survey (GSHS) data for identifying correlates of bullying to provide quality evidence for planning effective anti-bullying initiatives in Nepal.

## Materials and methods

### Data source

This study used nationally representative data from GSHS 2015, which was carried out to assess the health behaviors and factors associated with major causes of death and morbidity among school-going adolescents in Nepal. GSHS adopted a two-stage cluster sampling design and was conducted among 7–11 grades school adolescents. Initially, 74 eligible schools were selected from all the government and private schools of Nepal containing any of 7–11 grades. based on the probability proportional to school enrollment size. Classrooms were randomly selected based on the random number that was already assigned for each class. All the students (a total of 8670 students) from the sampled classes were eligible to participate in this study. Those students participated who got written consent from their parents. Non-response and selection probabilities were adjusted by determining the weighting factor of each student. Among 74 sampled schools; 68 schools (92%) participated, four were found to be closed during the data collection period, one could not be reached because of road obstruction due to flooding and one declined to participate. Altogether, 6531 students (75%) from sampled classes completed the GSHS questionnaire. The overall response rate was 69%. Out of these responses, 6529 questionnaires were usable after data cleaning. GSHS reported data of 6529 students from grade 7 to 11 of 68 schools from all the ecological regions of Nepal and this study analyzed all these data from GSHS dataset. A detail description of GSHS 2015 methods is reported elsewhere [29].

### Survey instrument

Nepal GSHS used a standard set of self-administered questionnaires which contained 91 questions; 58 core questions and 31 expanded questions on ten core modules: demographics, dietary behaviors, hygiene, violence and unintentional injury, mental health, tobacco use, alcohol and drug use, sexual behaviors, physical activity and protective factors. The questionnaire was translated into Nepali and back-translated into English to ensure the translation validity and also pre-testing was done to make the necessary modification in the questionnaire.

Research assistants (public health and nursing graduates) were recruited for data collection and trained on GSHS manual (provided by CDC) before moving to the field. All the students were provided with the information sheet and written consent form on the previous day of data collection and only those students participated who had obtained written consent form from parents/guardians.

### Study variables

The primary outcome of interest in our study is the victimization of bullying. This variable was measured using the following question: "During the past 30 days, on how many days were you bullied?" The option ranged from 0 to 30 days. For our analysis, we considered the response of one or more days to have been bullied.

The following are the independent variables used in this study (Table 1). Age was categorized and re-coded into early adolescent (11 to 14 years) and late adolescent (15 years and above) and other variables having Likert scale responses were re-coded into binary dichotomous variables; either 0 or 1 for logistic regression analysis.

### Data analysis

Complex sample analysis was carried out using the primary sampling unit, stratum and sample weight in STATA version 13.1. A p-value <0.05 was considered statistically significant for all the statistical analyses.

**Table 1. Description of Independent variables used in the study.**

| Variable | Questions | Code |
|---|---|---|
| Age | How old are you? | 1 = 11 to 14 years (Early adolescent) |
| | | 2 = 15 years and above (Late adolescent) |
| Sex | What is your sex? | 1 = Male |
| | | 2 = Female |
| Felt lonely | During the past 12 months, how often have you felt lonely? | 1 = Sometimes, most of the time, always |
| | | 0 = Never, rarely |
| Anxiety | During the past 12 months, how often have you been so worried about something that you could not sleep at night? | 1 = Sometimes, most of the time, always |
| | | 0 = Never, rarely |
| Considered suicide | During the past 12 months, did you ever seriously consider attempting suicide? | 1 = Yes |
| | | 2 = No |
| Attempted suicide | During the past 12 months, how many times did you actually attempt suicide? | 1 = 1 or more times |
| | | 0 = 0 times |
| Involved in physical fight | During the past 12 months, how many times were you in a physical fight? | 1 = 1 or more times |
| | | 0 = 0 times |
| Did not go to school due to unsafe | During the past 30 days, on how many days did you not go to school because you felt you would be unsafe at school or on your way to or from school? | 1 = 1 or more days |
| | | 0 = 0 days |
| Missed school without permission (Truancy) | During the past 30 days, on how many days did you miss classes or school without permission? | 1 = 1 or more days |
| | | 0 = 0 days |
| Smoking | During the past 30 days, on how many days did you smoke cigarettes? | 1 = 1 or more days |
| | | 0 = 0 days |
| Smokeless tobacco use | During the past 30 days, on how many days did you use any tobacco products other than cigarettes, such as chewing tobacco surti, khaini, gutka, or parag? | 1 = 1 or more days |
| | | 0 = 0 days |
| Alcohol use | During the past 30 days, on how many days did you have at least one drink containing alcohol? | 1 = 1 or more days |
| | | 0 = 0 days |
| Overweight | How tall are you without your shoes on? | >+1SD from the median for BMI by age and sex |
| | How much do you weigh without your shoes on? | |
| Underweight | How tall are you without your shoes on? | <-2SD from the median for BMI by age and sex |
| | How much do you weigh without your shoes on? | |
| Sexual risk behavior | During your life, with how many people have you had sexual intercourse? | 1 = 2 or more people |
| | | 0 = never had sexual intercourse, 1 person |
| Physically active | During the past 7 days, on how many days were you physically active for a total of at least 60 minutes per day? | 1 = 7 days |
| | | 0 = 0 to 6 days |
| Bullied | During the past 30 days, on how many days were you bullied? | 1 = 1 or more times |
| | | 0 = 0 times |

Pearson Chi-square test was used for bivariate analysis. We reported adjusted odds ratio obtained from multivariable logistic regression analysis to estimate the association between bullying and its health risk behavior and mental health behavior correlates.

## Results

More than half of the participants were of the age group 11 to 14 years (56.1%). Around half of the students were bullied at school (50.7%) with high prevalence among male (55.7%). Around

one-third of the participants had felt lonely (33.6%) and had anxiety (34.2%). Similarly, 13.6% had considered suicide and 10.3% had attempted suicide. Absenteeism due to unsafe feeling was reported among 41.2% of the students and 28.3% had missed school without taking permission. In the past 30 days, around 6.2% had smoked, 5.8% had used tobacco and 5.3% had used alcohol. Similarly, 39.2% were involved in physical fight and around 15.2% were physically active (Table 2).

### Mental health behaviors

Students facing loneliness (61.52%, 95% CI: 56.74, 66.08), anxiety (64.97%, 95% CI: 60.44, 69.25), considered suicide (60.8%, 95% CI: 54.95, 66.44), attempted suicide (72.9%, 95% CI: 67.24, 77.9), missed school feeling unsafe (61.66%, 95% CI: 57.15, 65.98), missed school without permission (60.81%, 95% CI: 56.38, 65.07), and involvement in physical fight (70.14%, 95% CI: 65.78, 74.17) had high prevalence of being bullied (Table 3).

### Health risk behaviors

Bullying prevalence was high among students who were involved in health risk behaviors like smoking (75.22%, 95%CI: 67.81, 81.39), using smokeless tobacco products (76.05%, 95%CI: 66.72, 83.42), drink alcohol (72.51%, 95%CI: 60.68, 81.84), and involved in sexual risk behavior (65.14%, 95%CI: 54.95, 74.12). Other variables like age, overweight, underweight, physically active were not found to have statistically significant association with bullying (Table 3).

Multivariable analysis revealed that a significant association was seen between bullied students and tobacco use (aOR 2.05, 95% CI: 1.15, 3.65) and involvement in physical fight (aOR 3.64, 95% CI: 2.94, 4.51). No clear associations were found between bullying and other risk variables like smoking, alcohol use, overweight, underweight, sexual risk behavior and physically active (Table 4).

Multivariable analysis showed that there was signification association between bullying and loneliness (aOR 1.36, 95% CI: 1.12, 1.64), anxiety (aOR 2.04, 95% CI: 1.65, 2.52), attempted suicide (aOR 2.08, 95% CI: 1.54, 2.81), missed school feeling unsafe (aOR 1.72, 95% CI: 1.34, 2.21) and missed school without permission (aOR 1.48, 95% CI: 1.17, 1.87) (Table 5).

## Discussion

This is a large scale study on school bullying in Nepalese school adolescents. In this study, we aimed to identify the factors associated with bullying behavior among adolescent students. We found significant association between bullying victimization and health risk behaviors like tobacco and involvement in a physical fight. Mental health problems like loneliness, anxiety, suicide attempt and school absenteeism were also significantly associated with bullying victimization.

Comparing the data of GSHS from the southeast Asian countries, Nepal reported the highest prevalence (51%) of bully-victims in the region followed by; Srilanka (37.9%), Timor Leste (28.3%), Thailand (27.8%), Bangladesh (24.6%), Indonesia (20.6%) and Myanmar (19.4%) [17] and demonstrated significant association with various risk factors. The finding in this study showed that males were more likely to become victims of a bully than females. One previous study analyzed the data of five cross-national surveys that were conducted among school aged children. The findings showed that, in four of the surveys, males were more likely to become victims of a bully and in one of the online survey; bullying prevalence was high among girls [30]. However, this finding might vary across schools, communities, countries, and cultures [9]. One of the possible explanations for this gender-wise variation might be that girls are more vulnerable to relational victimization like exclusion from groups, get threaten to damage

**Table 2. Characteristics of research participants.**

| | Male | | Female | | Total | |
|---|---|---|---|---|---|---|
| | N | %(95%CI) | N | %(95%CI) | N | %(95%CI) |
| **Age** | | | | | | |
| Early adolescent | 1452 | 53.3(50.2,56.4) | 1784 | 58.7(54,63.3) | 3236 | 56.1(52.7,59.4) |
| Late adolescent | 1551 | 46.7(43.6,49.8) | 1613 | 41.3(36.8,46) | 3164 | 43.9(40.6,47.3) |
| **Felt lonely** | | | | | | |
| Never | 1957 | 66.9(62.7,70.8) | 2144 | 65.9(63.3,68.4) | 4101 | 66.4(63.7,68.9) |
| Ever | 1009 | 33.1(29.2,37.3) | 1177 | 34.1(31.7,36.7) | 2186 | 33.6(31.1,36.3) |
| **Anxiety** | | | | | | |
| Never | 2039 | 68.2(64.4,71.8) | 2044 | 63.5(59.8,67.1) | 4083 | 65.8(62.6,68.9) |
| Ever | 963 | 31.8(28.2,35.6) | 1335 | 36.5(32.9,40.2) | 2298 | 34.2(31.1,37.4) |
| **Considered suicide** | | | | | | |
| No | 2611 | 86.7(83.3,89.6) | 2897 | 86.1(83,88.7) | 5508 | 86.4(83.8,88.6) |
| Yes | 343 | 13.3(10.4,16.7) | 438 | 13.9(11.3,17) | 781 | 13.6(11.4,16.2) |
| **Attempted suicide** | | | | | | |
| No | 2758 | 90.3(87.2,92.7) | 3061 | 89.1(86.2,91.5) | 5819 | 89.7(86.9,91.9) |
| Yes | 248 | 9.7(7.3,12.8) | 324 | 10.9(8.5,13.8) | 572 | 10.3(8.1,13.1) |
| **Involved in physical fight** | | | | | | |
| No | 1750 | 56.3(52.6,60) | 2239 | 65.2(60.9,69.2) | 3989 | 60.9(57.7,63.9) |
| Yes | 1260 | 43.7(40,47.4) | 1153 | 34.9(30.8,39.1) | 2413 | 39.2(36.1,42.3) |
| **Didn't go to school due to unsafe** | | | | | | |
| No | 1772 | 58(53.1,62.8) | 2122 | 59.6(52.7,66.2) | 3894 | 58.8(53.2,64.3) |
| Yes | 1215 | 42(37.2,46.9) | 1234 | 40.4(33.8,47.3) | 2449 | 41.2(35.7,46.8) |
| **Missed school without permission (Truancy)** | | | | | | |
| No | 2142 | 71.9(67.6,75.9) | 2392 | 71.5(66.6,75.9) | 4534 | 71.7(67.7,75.3) |
| Yes | 814 | 28.1(24.1,32.4) | 908 | 28.5(24.1,33.4) | 1722 | 28.3(24.7,32.3) |
| **Smoking** | | | | | | |
| No | 2703 | 91.1(88.5,93.1) | 3236 | 96.5(94.6,97.7) | 5939 | 93.8(92,95.3) |
| Yes | 251 | 8.9(6.9,11.5) | 91 | 3.5(2.3,5.4) | 342 | 6.2(4.7,8) |
| **Smokeless tobacco use** | | | | | | |
| No | 2803 | 92.4(89.9,94.4) | 3273 | 95.8(93.7,97.2) | 6076 | 94.2(92.1,95.7) |
| Yes | 205 | 7.6(5.6,10.1) | 110 | 4.2(2.8,6.3) | 315 | 5.8(4.3,7.9) |
| **Alcohol use** | | | | | | |
| No | 2765 | 92.9(91.1,94.4) | 3231 | 96.3(94.7,97.5) | 5996 | 94.7(93.1,95.9) |
| Yes | 191 | 7.1(5.6,8.9) | 98 | 3.7(2.5,5.3) | 289 | 5.3(4.1,6.9) |
| **Overweight** | | | | | | |
| No | 2847 | 93.4(90.1,95.6) | 3239 | 94.8(92.8,96.3) | 6086 | 94.1(91.7,95.9) |
| Yes | 169 | 6.6(4.4,9.9) | 167 | 5.2(3.7,7.2) | 336 | 5.9(4.1,8.3) |
| **Underweight** | | | | | | |
| No | 2309 | 86.2(82.5,89.2) | 2854 | 91.9(89.7,93.7) | 5163 | 89.1(86.4,91.4) |
| Yes | 384 | 13.8(10.8,17.5) | 244 | 8.1(6.3,10.3) | 628 | 10.9(8.6,13.6) |
| **Sexual risk** | | | | | | |
| No risk | 2734 | 95.1(93,96.6) | 3227 | 97.5(94.7,98.9) | 5961 | 96.4(94.3,97.7) |
| Risk | 153 | 4.9(3.4,7) | 66 | 2.5(1.1,5.3) | 219 | 3.6(2.3,5.7) |
| **Physically active** | | | | | | |
| No | 2430 | 82.6(77.4,86.8) | 2794 | 86.9(82.2,90.5) | 5224 | 84.8(80.5,88.2) |
| Yes | 554 | 17.4(13.2,22.6) | 536 | 13.1(9.5,17.8) | 1090 | 15.2(11.8,19.5) |
| **Bullying** | | | | | | |

*(Continued)*

**Table 2.** (Continued)

|  | Male | | Female | | Total | |
|---|---|---|---|---|---|---|
|  | N | %(95%CI) | N | %(95%CI) | N | %(95%CI) |
| No | 1334 | 44.3(39.9,48.9) | 1838 | 53.8(50,57.6) | 3172 | 49.3(45.6,53) |
| Yes | 1505 | 55.7(51.1,60.2) | 1427 | 46.2(42.4,50) | 2932 | 50.7(47,54.4) |

social relationship, manipulation, gossip and rumor-spread and thus, less likely to report the victimization compared to the boys.

## Mental health behaviors

As mental health behavior and bullying shows reciprocal relationship, we just aimed to study association between these two because of the cross-sectional nature of the study. This study revealed that victimization is significantly associated with loneliness and anxiety. In recent years, researchers have shown an increased interest in learning the relation between victimization with mental health outcomes and cognitive abilities in later life. Cohort studies from UK and USA reported a significant association of child victimization with persisting mental health problems across adulthood [6, 10]. Both perpetrators and victims are vulnerable to psychological health consequences. Perpetrators are more likely to become anti-social, involve in negative health behaviors and criminal activities in later life [17, 31, 32]. On the other hand, victims could be at increased risk of not only mental problems but also physical illnesses and injuries which may hamper the social development of a child.

There is also an increasing concern on assessing the root causes of harmful intentional acts these days. In this study, we found a significant association between suicide attempts and victimization. A systematic review showed that school bullying victims were 1.10 to 5.41 times more likely to show suicidal ideation and 2.45 to 2.76 times more likely to plan a suicide attempt as compared to non-bullied [33]. Another meta-analysis study, conducted with the reference of 47 studies, revealed that there is a positive association between bullying and suicidal behavior [34]. Physical victimization was associated with increased odds of suicide ideation and relational victimization was associated with suicidal attempt. Suicidal behaviour is also more prominent among cyberspace bullying victims [35, 36]. The increased use of the internet and social networking sites among school aged children has heightened the harmful consequences of cyber-bullying these days. Further work is required to establish causal links between cyber bullying and its mental and behavioural impact on school bullied victims in low-income countries.

Consistent with other literatures, this research found a positive association between bullied adolescents and absenteeism which was due to unsafe feeling and truancy than non-bullied [37, 38]. Several studies have shown that this fear and truancy due to bullying can lead to mental problems and lower academic achievement of students [15, 39]. Supporting this evidence, previous studies have emphasized the requirements of appropriate anti-bullying programs at schools [40–45]. It is therefore likely that learning outcomes could be improved through anti-school bullying strategies like school/classroom rules and policies, curriculum materials, parents/teachers trainings and meetings, disciplinary methods and improved playground supervision [46, 47].

## Health risk behaviors

Some of the previous studies have demonstrated the association between victimization and negative health behaviors (smoking, tobacco use, alcohol use, sexual risk behavior and physical

**Table 3. Prevalence of school bully-victims among school-going adolescents of Nepal.**

| Variables | Total Number | Weighted Count | Prevalence | 95%CI |
|---|---|---|---|---|
| **Age** | | | | |
| Early adolescent | 3130 | 1514 | 50.43 | 45.08,55.77 |
| Late adolescent | 3028 | 1453 | 51.45 | 47.54,55.35 |
| **Sex**\*\*\* | | | | |
| Male | 2839 | 1505 | 55.67 | 51.09,60.15 |
| Female | 3265 | 1427 | 46.17 | 42.4,49.98 |
| **Felt lonely**\*\*\* | | | | |
| Ever | 2102 | 1245 | 61.52 | 56.74,66.08 |
| Never | 3971 | 1671 | 45.38 | 41.4,49.43 |
| **Anxiety**\*\*\* | | | | |
| Ever | 2209 | 1349 | 64.97 | 60.44,69.25 |
| Never | 3955 | 1618 | 43.65 | 39.39,48.0 |
| **Considered suicide**\*\* | | | | |
| Yes | 752 | 455 | 60.84 | 54.95,66.44 |
| No | 5329 | 2460 | 49.02 | 44.71,53.35 |
| **Attempted suicide**\*\*\* | | | | |
| Yes | 544 | 391 | 72.9 | 67.24,77.9 |
| No | 5629 | 2587 | 48.47 | 44.59,52.36 |
| **Involved in physical fight**\*\*\* | | | | |
| Yes | 2339 | 1578 | 70.14 | 65.78,74.17 |
| No | 3845 | 1399 | 38.34 | 34.7,42.11 |
| **Missed school feeling unsafe**\*\*\* | | | | |
| Yes | 2345 | 1380 | 61.66 | 57.15,65.98 |
| No | 3784 | 1562 | 43.25 | 38.63,47.99 |
| **Missed school without permission**\*\*\* | | | | |
| Yes | 1644 | 971 | 60.81 | 56.38,65.07 |
| No | 4406 | 1942 | 46.91 | 42.73,51.13 |
| **Smoking**\*\*\* | | | | |
| Yes | 328 | 240 | 75.22 | 67.81,81.39 |
| No | 5740 | 2671 | 49.2 | 45.43,52.98 |
| **Smokeless tobacco use**\*\*\* | | | | |
| Yes | 290 | 217 | 76.05 | 66.72,83.42 |
| No | 5884 | 2756 | 49.41 | 45.79,53.04 |
| **Alcohol use**\*\*\* | | | | |
| Yes | 265 | 193 | 72.51 | 60.68,81.84 |
| No | 5814 | 2712 | 49.42 | 45.67,53.18 |
| **Overweight** | | | | |
| Yes | 315 | 162 | 53.15 | 43.39,62.67 |
| No | 5886 | 2826 | 50.72 | 47.22,54.22 |
| **Underweight** | | | | |
| Yes | 598 | 312 | 53.6 | 48.58,58.55 |
| No | 4910 | 2324 | 50.08 | 46.56,53.59 |
| **Sexual risk behavior**\*\* | | | | |
| Yes | 213 | 129 | 65.14 | 54.95,74.12 |
| No | 5769 | 2737 | 50.26 | 46.44,54.07 |
| **Physically active** | | | | |
| Yes | 1068 | 501 | 49.36 | 42.29,56.45 |

(*Continued*)

**Table 3.** (Continued)

| Variables | Total Number | Weighted Count | Prevalence | 95%CI |
|---|---|---|---|---|
| No | 5035 | 2440 | 51.19 | 47.36,55.01 |

*p < 0.05
**p < 0.01
***p < 0.001.

**Table 4. Multivariable analysis of health risk behaviors of school bully-victims among school-going adolescents of Nepal.**

| Variables | Crude OR | 95% CI | Adjusted OR* | 95% CI |
|---|---|---|---|---|
| **Smoking (Ref:No)** | | | | |
| Yes | 3.13 | 2.13,4.61 | 1.43 | 0.91,2.26 |
| **Smokeless tobacco use (Ref:No)** | | | | |
| Yes | 3.25 | 2.07,5.11 | 2.05 | 1.15,3.65 |
| **Alcohol use (Ref:No)** | | | | |
| Yes | 2.70 | 1.55,4.68 | 1.1 | 0.74,1.62 |
| **Overweight (Ref:No)** | | | | |
| Yes | 1.10 | 0.79,1.55 | 0.99 | 0.71,1.39 |
| **Underweight (Ref:No)** | | | | |
| Yes | 1.15 | 0.91,1.46 | 1.13 | 0.83,1.54 |
| **Sexual Risk Behavior (Ref:No)** | | | | |
| Yes | 1.85 | 1.20,2.85 | 0.87 | 0.51,1.47 |
| **Involved in physical fight (Ref:No)** | | | | |
| Yes | 3.78 | 3.06,4.66 | 3.64 | 2.94,4.51 |
| **Physically active (Ref:No)** | | | | |
| Yes | 0.93 | 0.70,1.24 | 0.96 | 0.69,1.34 |

*Adjusted for age, sex, smoking, smokeless tobacco use, alcohol use, overweight, underweight, sexual risk behavior, involved in physical fight and physically active.

**Table 5. Multivariable analysis of mental health behaviors of school bully -victims among school-going adolescents of Nepal.**

| Variables | Crude OR | 95% CI | Adjusted OR* | 95% CI |
|---|---|---|---|---|
| **Loneliness (Ref:Never)** | | | | |
| Ever | 1.92 | 1.58,2.34 | 1.36 | 1.12,1.64 |
| **Anxiety(Ref:Never)** | | | | |
| Ever | 2.39 | 1.93,2.96 | 2.04 | 1.65,2.52 |
| **Considered suicide(Ref:No)** | | | | |
| Yes | 1.62 | 1.16,2.26 | 0.94 | 0.7,1.24 |
| **Attempted suicide(Ref:No)** | | | | |
| Yes | 2.86 | 2.12,3.86 | 2.08 | 1.54,2.81 |
| **Missed school feeling unsafe(Ref:No)** | | | | |
| Yes | 2.11 | 1.67,2.67 | 1.72 | 1.34,2.21 |
| **Missed school without permission(Ref:No)** | | | | |
| Yes | 1.76 | 1.43,2.16 | 1.48 | 1.17,1.87 |

*Adjusted for age, sex, loneliness, anxiety, considered suicide, attempted suicide, missed school feeling unsafe and missed school without permission.

inactivity) [20, 21, 23, 48, 49]. Contrary to these studies, this study found no significant association of victimization with risk behaviors like smoking, alcohol use, sexual risk behavior and physical inactivity. However in this study, we found that bully-victims were more susceptible to engage in smokeless tobacco use. This study has been unable to demonstrate whether the result was influenced by other forms of bullying or was based on never tried tobacco students. Further studies need to be undertaken to investigate the direction and possibility of these risk variables.

Consistent with other studies, this study showed a significant association between victimization and being involved physical fight [22, 23]. On one hand, some bully-victims mentally harm themselves as discussed above and some others choose fighting back which may lead to serious injuries. On other hand, there is also the possibility that adolescents who were already engaged in a physical fight become the targets of bullying. Also, previous studies have reported the linkage of bullying with abnormal body weight (overweight, underweight) [24, 25]. Evidence from these studies showed that overweight adolescents were more likely to experience verbal torture whereas underweight adolescents were more likely to experience physical and relational victimization. Contrary to that, the result of this study showed no association with weight. The reason for these inconsistencies might be due to the limitations of this study. There is a need for further studies to clarify the full picture. Therefore, this component is an important recommendation for future research.

There are some limitations in this study. As GSHS was designed to assess the behavioral and protective factors of 10 different key areas, due to its broad nature, bullying tends to occupy small space with very few questions in the survey. Also, being a cross-sectional study design, associated factors might have limited implications for the casualty. Because of this, we couldn't detect whether bully-victims showed harmful mental and health risk behaviors or adolescents already having this behavior were becoming the victims of bullying. Also, the participation of school students alone does not represent the adolescents of the whole nation. The findings could differ between school enrolled and non-enrolled adolescents. This survey lacks the data based on geographical and socio-economic information, schools' characteristics (government or private schools) and other factors that might determine bully. As this is self-reported data, there might be some potential biases regarding the response of adolescents. Also, it does not explain detail on the nature of bullying such as perpetrator characteristics, the role of cyber-bullying etc. Despite these limitations, this study is the representation of a large sample of Nepalese students and has significant implications on adolescent development.

## Conclusion

This study identified school bullying as one of the significant public health issues in Nepal. This study showed that school bullying is associated with poor mental and health risk behaviors that have a negative impact on present and/or later life. Despite the cross-sectional nature of the study, the findings of this study provide valuable insights for better understanding of correlates of school bullying in Nepal. Ensuring safe school and family environment should be the main concern while planning anti-bullying strategies in context of Nepal. The combined effort from parents, teachers, school administrators and school staff is essential in preventing school bullying.

## Supporting information

**S1 Table. Description of Independent variables used in the study.**
(PDF)

**S2 Table. Characteristics of research participants.**
(PDF)

**S3 Table. Prevalence of school bully-victims among school going adolescents of Nepal.**
(PDF)

**S4 Table. Multivariable analysis of health risk behaviors of school bully-victims among school going adolescents of Nepal.**
(PDF)

**S5 Table. Multivariable analysis of mental health behaviors of school bully -victims among school going adolescents of Nepal.**
(PDF)

## Author Contributions

**Conceptualization:** Tamanna Neupane, Achyut Raj Pandey, Bihungum Bista, Binaya Chalise.

**Formal analysis:** Tamanna Neupane, Achyut Raj Pandey, Bihungum Bista, Binaya Chalise.

**Writing – original draft:** Tamanna Neupane.

**Writing – review & editing:** Achyut Raj Pandey, Bihungum Bista, Binaya Chalise.

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
