## [Decision Letter · Decision Letter 0]

5 Jun 2020

PONE-D-20-12333

Correlates of bullying victimization among school adolescents in Nepal: Findings from 2015 Global School-Based Student Health Survey Nepal

PLOS ONE

Dear Dr. Neupane,

Thank you for submitting your manuscript to PLOS ONE. After careful consideration, we feel that it has merit but does not fully meet PLOS ONE’s publication criteria as it currently stands. Therefore, we invite you to submit a revised version of the manuscript that addresses the points raised during the review process.

We look forward to receiving your revised manuscript.

Kind regards,

Pranil Man Singh Pradhan

Academic Editor

PLOS ONE

Journal Requirements:

"The authors have received no funding for the publication of this study."

We note that one or more of the authors are employed by a commercial company: Abt Associates.

2.1. Please provide an amended Funding Statement declaring this commercial affiliation, as well as a statement regarding the Role of Funders in your study. If the funding organization did not play a role in the study design, data collection and analysis, decision to publish, or preparation of the manuscript and only provided financial support in the form of authors' salaries and/or research materials, please review your statements relating to the author contributions, and ensure you have specifically and accurately indicated the role(s) that these authors had in your study. You can update author roles in the Author Contributions section of the online submission form.

2.2. Please also provide an updated Competing Interests Statement declaring this commercial affiliation along with any other relevant declarations relating to employment, consultancy, patents, products in development, or marketed products, etc. 

Reviewers' comments:

Reviewer's Responses to Questions

**Comments to the Author**

1. Is the manuscript technically sound, and do the data support the conclusions?

Reviewer #1: Yes

Reviewer #2: Yes

2. Has the statistical analysis been performed appropriately and rigorously? 

Reviewer #1: No

Reviewer #2: Yes

3. Have the authors made all data underlying the findings in their manuscript fully available?

Reviewer #1: No

Reviewer #2: Yes

4. Is the manuscript presented in an intelligible fashion and written in standard English?

Reviewer #1: Yes

Reviewer #2: Yes

5. Review Comments to the Author

Reviewer #1: This is a well written article highlighting a very important area of school health on bullying. However, there are certain aspects that could be improved in the revision to reach a publishable quality.

Major comments:

Methods:

The authors have mentioned that they have used a nationally representative data. However, no information is provided on the regions from which the schools were selected. Were the selected schools government or private schools, were these schools selected across the country or confined to a certain area alone?

What were the characteristics of the schools? For a country like Nepal, schools' characteristics vary widely between regions as well as between sectors. The problems faced by students from one of the most expensive school in Kathmandu isn't comparable to the students who need to walk for hours to reach the school in some remote parts of the country. Thus, more information on the schools are necessary.

The authors did acknowledge in the limitations section that certain information about schools are missing. In that case, the authors should provide an elaborate explanation on how it is still a nationally representative data despite the lack of information in the methods section. Or provide more details about the schools.

The method section has elaborated about the school survey as a primary data collection study. However, as this is a study based on secondary data analysis, it would be important for the authors to also elaborate the methods on how the data was retrieved? Were all the data from the survey analysed? Was the sample size calculated for this secondary study? Were there any inclusion or exclusion criteria for selecting the data for this secondary analysis study? In short, please focus on the methods used for this part of the study rather than on the already published survey.

Results:

-It would have been better if Table 1. was about the general characteristics instead of the variables. The table on variables could have been added as a supplementary file.

-In Table 1. I see all the variables are dichotomized but for sensitive issues such as mental health (anxiety and loneliness), e.g. being anxious all the times, sometimes or most of the time have different implications. It looses information on the severity if always, sometimes and most of the time are grouped into one category. So either it should be grouped as ''ever'' and ''never''. Or it would have been better if the authors created dummy variables for each severity such as for anxiety- sometimes=1, rest =0, then always=1, rest=0, and so on to fit in each severity in the analysis. If its not possible, please acknowledge it as a limitation that dichotomizing the variables could have lead to potential loss of information and it doesn't reflect the severity.

-In the multivariable analyses, why weren't the demographic characteristics such as age and sex adjusted? They are important confounders and it is important to control them in multivariable analysis given that the sex had already shown some significant association with bullying.

-Ideally, if possible the schools or clusters of schools too should have been controlled in the analysis. As there are many schools, it is understandable that all schools cannot be controlled. Instead they could be grouped into similar schools such as government vs private schools or schools by region and controlled accordingly because schools are important confounders affecting the results for the reasons mentioned above. If it is impossible for this study to control school please provide an explanation.

Minor comments

-Line 38: Please spell out DALYS when it appears for the first time in the manuscript

-Line 40: Please spell out SDI

I would recommend the authors to re-analyse the data addressing the comments above or provide justification.

Reviewer #2: The overall manuscript seems technically sound. The keywords are missing in the abstract which should have been mentioned. The manuscript has been prepared in an intelligible fashion and standard English has been used. But framing of sentences can be made more concise in overall manuscript so that the article is more interesting for readers. The statistical analysis has been performed appropriately and vigorously. Operational definitions for overweight and underweight not mentioned under variable description. Recommendations can be made more specific towards associated factors and variables.

6. PLOS authors have the option to publish the peer review history of their article (what does this mean?). If published, this will include your full peer review and any attached files.

Reviewer #1: No

Reviewer #2: No

---

## [Author Response · Author response to Decision Letter 0]

23 Jun 2020

Reviewer #1:

Methods:

Reviewer’s comment: The authors have mentioned that they have used a nationally representative data. However, no information is provided on the regions from which the schools were selected. Were the selected schools government or private schools, were these schools selected across the country or confined to a certain area alone?

What were the characteristics of the schools? For a country like Nepal, schools' characteristics vary widely between regions as well as between sectors. The problems faced by students from one of the most expensive school in Kathmandu isn't comparable to the students who need to walk for hours to reach the school in some remote parts of the country. Thus, more information on the schools are necessary.

The authors did acknowledge in the limitations section that certain information about schools are missing. In that case, the authors should provide an elaborate explanation on how it is still a nationally representative data despite the lack of information in the methods section. Or provide more details about the schools.

The method section has elaborated about the school survey as a primary data collection study. However, as this is a study based on secondary data analysis, it would be important for the authors to also elaborate the methods on how the data was retrieved? Were all the data from the survey analysed? Was the sample size calculated for this secondary study? Were there any inclusion or exclusion criteria for selecting the data for this secondary analysis study? In short, please focus on the methods used for this part of the study rather than on the already published survey.

Author’s response: This study is secondary data analysis of Global School based Health Survey Nepal and data set of this survey can be accessed from WHO website (https://www.who.int/ncds/surveillance/gshs/nepaldataset/en/). This study includes all the data of the data set, so, no any separate sample size was calculated especially for this study. In the first stage of sampling, all the schools containing any of 7-11 class across the country were listed and 74 schools were selected with probability proportional to school enrolment size. According to the researchers involved in the study, schools from both government and private schools were included in the survey but GSHS report hasn’t mentioned about it. Similarly, while drawing schools, ecological belt (hill, mountain and terai) was also considered. From each ecological belt, 25 schools were taken. However, the dataset doesn’t contain information on it.

Results:

Reviewer’s comment: It would have been better if Table 1. was about the general characteristics instead of the variables. The table on variables could have been added as a supplementary file.

In Table 1. I see all the variables are dichotomized but for sensitive issues such as mental health (anxiety and loneliness), e.g. being anxious all the times, sometimes or most of the time have different implications. It loses information on the severity if always, sometimes and most of the time are grouped into one category. So either it should be grouped as ''ever'' and ''never''. Or it would have been better if the authors created dummy variables for each severity such as for anxiety- sometimes=1, rest =0, then always=1, rest=0, and so on to fit in each severity in the analysis. If it’s not possible, please acknowledge it as a limitation that dichotomizing the variables could have lead to potential loss of information and it doesn't reflect the severity.

Author’s response: Revised as suggested (included characteristics of participants, coded mental factors as ever and never) 

Reviewer’s comment: In the multivariable analyses, why weren't the demographic characteristics such as age and sex adjusted? They are important confounders and it is important to control them in multivariable analysis given that the sex had already shown some significant association with bullying.

Ideally, if possible the schools or clusters of schools too should have been controlled in the analysis. As there are many schools, it is understandable that all schools cannot be controlled. Instead they could be grouped into similar schools such as government vs private schools or schools by region and controlled accordingly because schools are important confounders affecting the results for the reasons mentioned above. If it is impossible for this study to control school please provide an explanation.

Author’s response: Regarding controlling school related features (public vs private, ecological belts) in multivariable analysis; the data set do not provide the information about the type of schools and ecological belts in which schools were located. So, this make impossible to adjust school related features. We adjusted age and sex in multivariable analysis.

Other Minor comments

Reviewer’s comment: Line 38: Please spell out DALYS when it appears for the first time in the manuscript

Author’s response: Revised as suggested

Reviewer’s comment: Line 40: Please spell out SDI

Author’s response: Revised as suggested

Reviewer #2: 

Reviewer’s comment: The overall manuscript seems technically sound. The keywords are missing in the abstract which should have been mentioned. The manuscript has been prepared in an intelligible fashion and standard English has been used. But framing of sentences can be made more concise in overall manuscript so that the article is more interesting for readers. The statistical analysis has been performed appropriately and vigorously. Operational definitions for overweight and underweight not mentioned under variable description. Recommendations can be made more specific towards associated factors and variables.

Author’s response: Revised as suggested 

Operational definitions for overweight: >+1SD from median for BMI by age and sex

Operational definitions for underweight: <-2SD from median for BMI by age and sex

Funding statement: Abt Associates Pty Ltd (Nepal Office) provided support in the form of salary for author Achyut Raj Pandey. There are no patents, products in development or marketed products associated with this research to declare. This does not alter our adherence to PLOS ONE policies on sharing data and materials

Competing interests statement. Abt Associates Pty Ltd (Nepal office) provided support in the form of salary for author Achyut Raj Pandey but did not have any additional role in the study design, data collection and analysis, decision to publish or preparation of the manuscript. The specific roles of authors is articulated in the authors contributions, section

---

## [Decision Letter · Decision Letter 1]

17 Jul 2020

PONE-D-20-12333R1

Correlates of bullying victimization among school adolescents in Nepal: Findings from 2015 Global School-Based Student Health Survey Nepal

PLOS ONE

Dear Dr. Neupane,

Thank you for submitting your manuscript to PLOS ONE. After careful consideration, we feel that it has merit but does not fully meet PLOS ONE’s publication criteria as it currently stands. Therefore, we invite you to submit a revised version of the manuscript that addresses the points raised during the review process.

We look forward to receiving your revised manuscript.

Kind regards,

Pranil Man Singh Pradhan

Academic Editor

PLOS ONE

Reviewers' comments:

Reviewer's Responses to Questions

**Comments to the Author**

1. If the authors have adequately addressed your comments raised in a previous round of review and you feel that this manuscript is now acceptable for publication, you may indicate that here to bypass the “Comments to the Author” section, enter your conflict of interest statement in the “Confidential to Editor” section, and submit your "Accept" recommendation.

Reviewer #1: All comments have been addressed

Reviewer #2: All comments have been addressed

2. Is the manuscript technically sound, and do the data support the conclusions?

Reviewer #1: Yes

Reviewer #2: Yes

3. Has the statistical analysis been performed appropriately and rigorously? 

Reviewer #1: Yes

Reviewer #2: Yes

4. Have the authors made all data underlying the findings in their manuscript fully available?

Reviewer #1: No

Reviewer #2: Yes

5. Is the manuscript presented in an intelligible fashion and written in standard English?

Reviewer #1: Yes

Reviewer #2: Yes

6. Review Comments to the Author

Reviewer #1: The authors have addressed most of the earlier comments adequately.

However, it would be better if the authors could also include some of the responses to the comments as part of the manuscript in the limitations section. Such as the authors mentioned that "According to the researchers involved in the

study, schools from both government and private schools were included in the survey

but GSHS report hasn’t mentioned about it."

This statements could be included (not entirely but briefly and rephrased) in the limitation section.

The authors have discussed the key findings elaborately with reference to global literature on bullying. However, it would be insightful if the authors could also elaborate a little on potential solutions to address bullying in schools for a country like Nepal. The authors concluded that "the findings of this study provide valuable insights for planning an effective approach in preventing bullying at schools". It would be insightful to understand what these "effective approaches" could be. A paragraph focusing on recommendations could help this paper provide more depth given that the paper uses a nationally representative data.

The manuscript needs proofreading.

Lines 39-42- The sentence is too long and difficult to follow. It can be broken down into two different sentences. Also "in" is missing before 2017 (line 41).

There are prepositions missing in some other parts of the manuscript as well.

Reviewer #2: The suggested revisions has been addressed. As the research is an analysis of a secondary data the limitations of selection of study population and sample still remains. Lack of informations about variations of schools also cannot be described more. But the research has been able to highlight a sensitive issue concerning mental health issues among school going adolescents of developing country like Nepal. Hence, the research should be encouraged to disseminate its findings in order to contribute to further research and interventions by concerned authorities.

7. PLOS authors have the option to publish the peer review history of their article (what does this mean?). If published, this will include your full peer review and any attached files.

Reviewer #1: No

Reviewer #2: No

---

## [Author Response · Author response to Decision Letter 1]

21 Jul 2020

Reviewer’s comment: Have the authors made all data underlying the findings in their manuscript fully available?

Author’s response: The dataset supporting the conclusions of this article is open access data and is available in the WHO website: https://www.who.int/ncds/surveillance/gshs/nepaldataset/en/

Reviewer’s comment: it would be better if the authors could also include some of the responses to the comments as part of the manuscript in the limitations section. Such as the authors mentioned that "According to the researchers involved in the study, schools from both government and private schools were included in the survey but GSHS report hasn’t mentioned about it." This statements could be included (not entirely but briefly and rephrased) in the limitation section.

Author’s response: Included as suggested in limitation section (This survey lacks the data based on geographical and socio-economic information, schools’ characteristics (government or private schools) and other factors that might determine bully.)

Reviewer’s comment: The authors have discussed the key findings elaborately with reference to global literature on bullying. However, it would be insightful if the authors could also elaborate a little on potential solutions to address bullying in schools for a country like Nepal. The authors concluded that "the findings of this study provide valuable insights for planning an effective approach in preventing bullying at schools". It would be insightful to understand what these "effective approaches" could be. A paragraph focusing on recommendations could help this paper provide more depth given that the paper uses a nationally representative data.

Author’s response: Revised as suggested in conclusion section

Reviewer’s comment:

The manuscript needs proofreading.

Lines 39-42- The sentence is too long and difficult to follow. It can be broken down into two different sentences. Also "in" is missing before 2017 (line 41).

There are prepositions missing in some other parts of the manuscript as well.

Author’s response: 

Revised as suggested. Proofreading is also done on the manuscript.

---

## [Editor Report · Decision Letter 2]

27 Jul 2020

Correlates of bullying victimization among school adolescents in Nepal: Findings from 2015 Global School-Based Student Health Survey Nepal

PONE-D-20-12333R2

Dear Dr. Neupane,

We’re pleased to inform you that your manuscript has been judged scientifically suitable for publication and will be formally accepted for publication once it meets all outstanding technical requirements.

Kind regards,

Dr. Pranil Man Singh Pradhan

Academic Editor

PLOS ONE
---

## [Editor Report · Acceptance letter]

3 Aug 2020

PONE-D-20-12333R2 

Correlates of bullying victimization among school adolescents in Nepal: Findings from 2015 Global School-Based Student Health Survey Nepal 

Dear Dr. Neupane:

I'm pleased to inform you that your manuscript has been deemed suitable for publication in PLOS ONE. Congratulations! Your manuscript is now with our production department. 

Kind regards, 

on behalf of

Dr. Pranil Man Singh Pradhan 

Academic Editor

PLOS ONE